# The Mimosa Manifesto

Version: March 2021

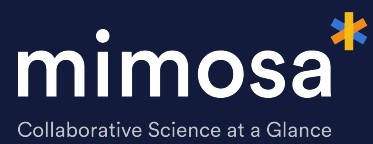

**Survey the state of your field. Share your research as you go. Get fast feedback. Find like-minded collaborators.**

## Prelude

Project by Lana Sinapayen, Sony Computer Science Laboratories Kyoto.
Mimosa is a platform for open collaboration in science. It is currently in development, deployment planned for 2021. This document presents an ideal version of Mimosa as it would be after release.

Invision design prototype: https://bit.ly/3t4RJNk
Google doc for comments: https://bit.ly/3cgEB0K
TL;DR in 7 slides: https://bit.ly/30wXwi8
Design description for the visually impaired: https://bit.ly/3sZDHME

## Science is a debate.

Debates happen where there is wiggle room for interpretation. There is no debate when all parties agree, or when all parties know why they disagree.

**Scientific debates can be settled by agreeing on an experimental protocol**. Good protocols identify wiggle room and preemptively get rid of it, by fixing the interpretation of experimental results before the experiment proceeds. "Are doctors transmitting deadly illnesses from cadavers to birthing mothers? Have some doctors wash their hands after autopsies. Let us agree that if their patients have better survival rates than usual, it means that infections travel on the hands of doctors (see Carter 1985)."
    Experimental results might tell you which way the settlement goes, but ideally the debate itself ends with the protocol. From this point of view, Science is the art of defining convincing protocols: scientific papers are often more interesting and more rigorous when they are written by two people who start out genuinely disagreeing.

Mimosa is an attempt at harnessing both support and disagreement in science into a productive, collaborative format. Mimosa also tries to address many of the numerous recognised issues within the current format for sharing science, born at a different time and for the wrong reasons.

When it first started, Wikipedia was greeted with suspicion. It is now a major platform for finding information, used by all demographics. Wikipedia has a famous rule: "No original research."

**Mimosa aspires to be that free, open-collaborative online platform created and maintained by a community of volunteer contributors, dedicated to original research.**

## Who can use Mimosa?

Mimosa fosters. . .

*[education, citizen science]*

**A 10 year-old** logs into the private Mimosa group set up by her teacher. She selects one of the proposed questions: "Do green beans grow faster in the sun or in the shadow?" and finds the experiment idea that her group collectively drafted the previous week. She clicks "add data" and starts inputting the height of her 4 potted beans.

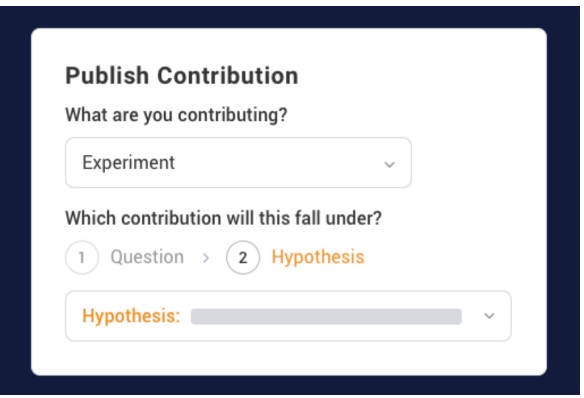

*[serendipity, cross-field cooperation]*

**A data analyst** working on vaccine microparticles looks up the tag "microparticles" and stumbles upon a dataset showing microparticle concentration in the air of US cities. The analysis appended to the dataset concludes that air pollution is not correlated with public transportation availability. The data analyst disagrees: he appends his own analysis to the same dataset, and proposes edits to the experimental protocol.

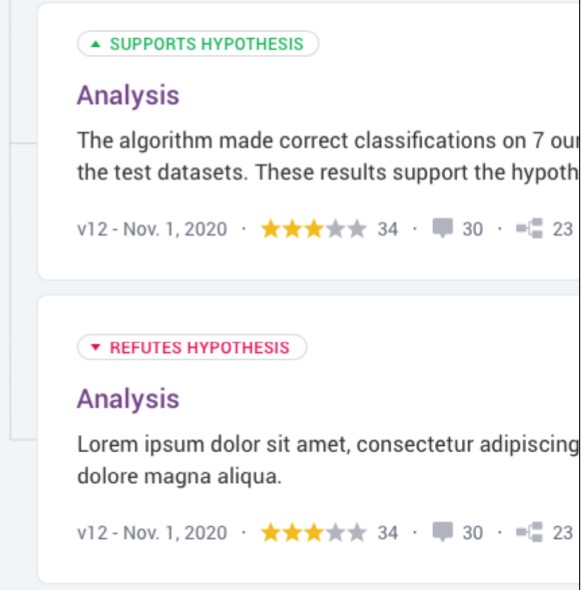

**A conference panel moderator** shares Mimosa links to the topics of the day with the audience. As the discussion progresses, arguments and counter-arguments are logged in the platform by the moderator. The audience rates and comments the content in real time, and the moderator picks some comments to ask the panelists. At the end of the panel session, the discussion can continue asynchronously and everyone can see where the answers stand.

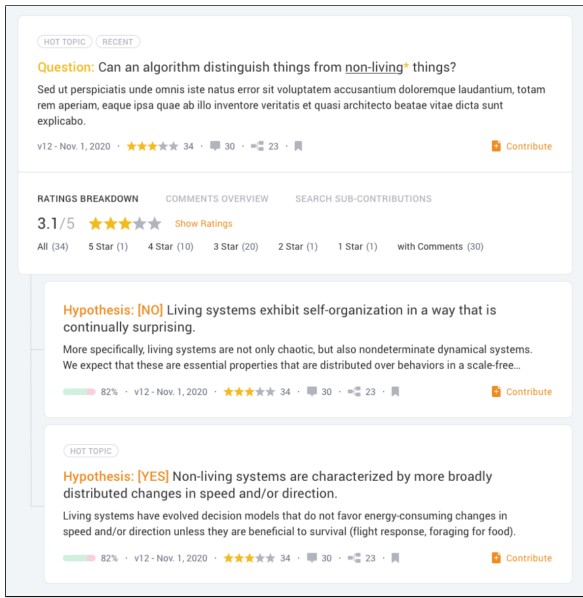

*[open questions discovery]*

**A graduate student** is looking for a topic to write her research proposal. She sees that the conditions of Open Ended Evolution are hotly debated, with many hypotheses having as many supporting analyses as refuting analyses, and under-explored experimental data. She decides to settle the question.

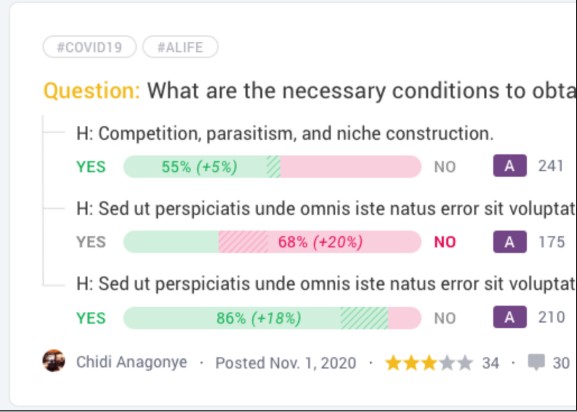

*[consensus summary for non-specialists]*

**A journalist** is looking for a definitive answer about extraterrestrial life. From mimosa's analytics, he can see that there is no scientific consensus, and decides to interview some of the highest rated contributors to the question. After writing his article, he submits a short meta-analysis through Mimosa that his sources can vet.

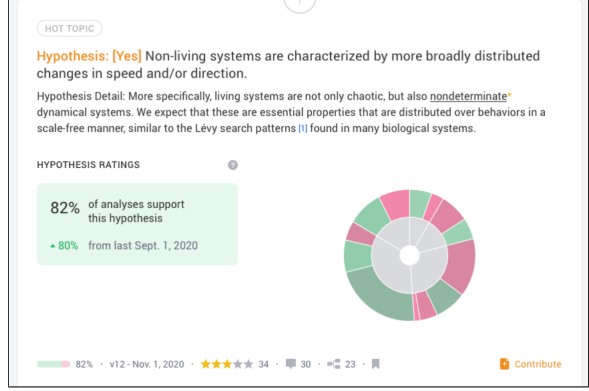

[rare data aggregation]

**A sufferer from the rare "visual snow" syndrome** notices that their symptoms get better when staring at blue light. They decide to publish this finding as a new research question on Mimosa, hoping to attract the interest of clinical researchers.

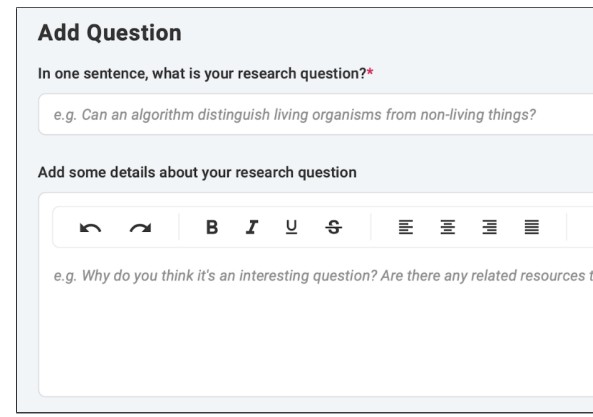

*[hiring]*

**A Principal Investigator** is looking to hire a postdoc. She has had good interactions with a frequent commenter on her publications, and sees from his profile that he is a highly rated hypothesis contributor, and has good records on his experimental protocols. She offers him a position.

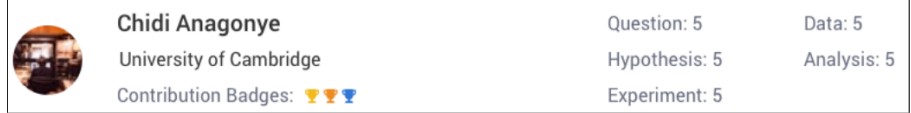

# Principles guiding the implementation of Mimosa

Through use cases, the previous section presented functions that improve the interactions between people and science. This section focuses on how Mimosa intends to improve science itself. The concept of a single "paper" does not exist in Mimosa: in the following, "paper" refers to the current concept of scientific paper published in a journal or an archive, while "contribution" is the minimal unit of publication in Mimosa.

Keys

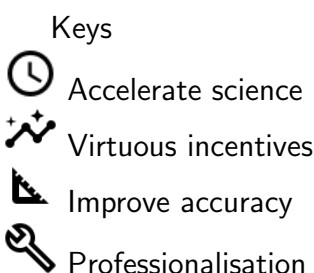

🕐 Accelerate science

Virtuous incentives

Improve accuracy

Professionalisation

## Micro-Publication

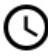

Traditionally, research is done in secrecy until a polished manuscript can be published. Mimosa

subscribes to the publish-as-you-go model. A contribution tree organizes contributions of different types: Research Question, Hypothesis, Experimental Protocol, Experimental Data, and Analysis. Contributions are added as the research progresses. The tree can have multiple branches, created by one or several contributors.

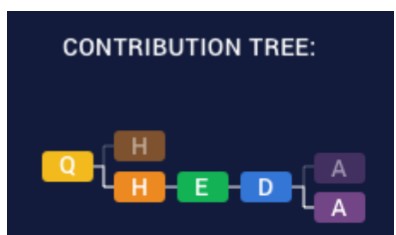

## Incremental improvement through feedback

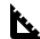

After publication, every type of contribution can receive comments and ratings (for example, hypotheses are rated on clarity and refutability). Updates to the contribution are tracked through version control. Since Experimental Protocols can be published before running the experiment, Mimosa effectively supports not only pre-registration, but also external review of the protocol before the experiment is conducted. Except for harassment and insults, contributions can never be rejected, only downvoted and/or improved.

## Virtuous incentives

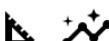

The research world is full of perverse incentives and bad behavior. Mimosa implements several safeguards. **1. Show me who you are**: a user's most flagged (inappropriate) comments are displayed publicly on their profile page for two weeks. **2. Right to improve**: a user's most recent ratings are weighted higher than their older ratings, so that people have an incentive to improve but also to stay good. **3. Ubiquity**: a user's activity (publications and reviews) outside of the Mimosa platform can be rated in the platform if they so desire and if that activity is claimed through their ORCID profile. **4. Responsible anonymity**: a user can contribute and comment anonymously, but the ratings on their anonymous activity will still be integrated to their public ratings. This protects from retaliation while still encouraging fair interactions. Anonymity can be invoked or revoked by the user at any time.

## Reciprocity

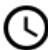

Any contribution can be reviewed, but contributors can flag their contributions to be reviewed in

priority, if they have themselves written at least 3 reviews per request. This is a way to attract attention to one's work, while ensuring that one gives back to the community.

## Inclusivity

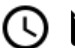

Not all stages of all sciences require higher education and big grants, but all of science benefits from wider point of views. Mimosa's barrier to entry is low. The platform is free, value judgements are prominent and explorable, and clarity is enforced through the structure of the contribution tree. People from different backgrounds can easily see if a claim is disputed or widely accepted, and contribute according to their ability. Mimosa will conform to the WCAG2.1 guidelines at level A.

## Professionalisation of roles

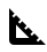 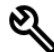

Scientists have always been expected to play many roles in the research process: researcher, data analyst, technician, statistician, illustrator, manuscript editor, reviewer, public speaker, publicist, team manager, mentor. . . Few if any scientists are truly good at every single one of these many jobs. Mimosa's reputation system allows users to discover what they are good at, and to find help for those tasks they are less confident with. All mature professions go through a professionalisation stage, and it might be time for science to do the same.

## Salient deprecation

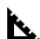

All hypotheses have a score indicating how strongly the hypothesis is supported or refuted by analyses of experimental data. This score is calculated from users ratings of the analyses, weighted by how well the rating users themselves are rated as reviewers. Hypotheses linked inside Mimosa through inline citations or embedded in external websites are displayed with their score: it is immediately noticeable if an argument is faulty or based on since-refuted hypotheses.

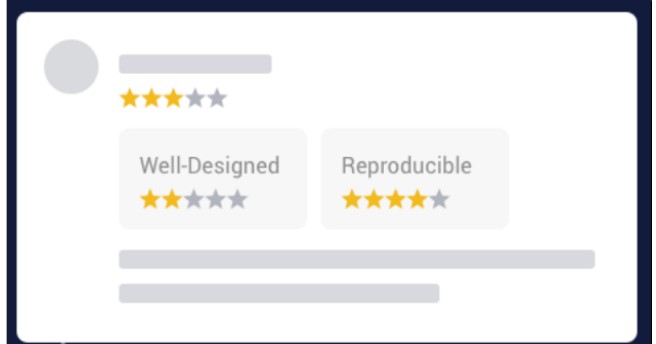 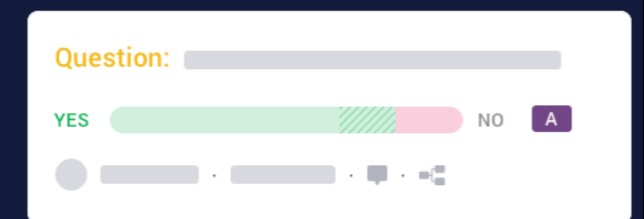

## Replication

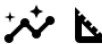

A research protocol that has been improved through feedback and vetted should attract replications. To facilitate this incentive, all experimental data based on a given protocol, no matter which user contributes it and when, is accessible on the same level of the contribution tree in Mimosa. There is no need to write a whole new paper, as all necessary information (from research question to protocol) is accessible in the same tree as the replication experiment.

## Funding

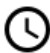

Eventually, researchers will be able to call for funding through the platform, in the same way as on Patreon or Kickstarter. Individuals may contribute amounts as small as $1, companies and governments may award bigger grants. There will be no need to write grant applications on the platform, as the content will already be public and polished through community interaction. Mimosa will retain a minimal percentage of the funds, only necessary to pay for server and database fees.

## Obsolescence of journals and conferences

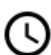

People use what is convenient and socially approved. If Mimosa reaches social acceptance, and manages to provide both prestige and employment opportunities, there will be less incentive to go through the pain and inconveniences of publishing in a journal. Conferences should also slowly be replaced by focus groups: people who have been collaborating on the same research question on Mimosa will naturally want to organise regular live discussions, and the demand for bigger annual conferences might disappear.

## Open Access, Open Source

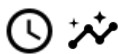

Mimosa lowers the barrier for non-scientists to consume and contribute to science. Mimosa will not charge for publication or access. In addition, the code of the platform will be public on GitHub and the data will be accessible through an open API to encourage modifications, improvements, and the birth of similar platforms. Because the contents of the platform is dynamic and uses ORCID authentication, there is no limitation to user names. A user can publish under a pseudonym, change their first or last name as they need, not have a last name... User-generated content will be awarded Open Science Badges (Preregistration badge, Open Materials badge, Open Data badge, see Kidwell et al. 2016) if they respect open science standards; the badges will serve as filters, and be displayed prominently on the contributions.

# The review process

Anyone can comment on public contributions, propose revisions, and rate them on predefined scales. These 3 elements have the role currently played by peer review in journals: evaluate contributions on objective and subjective criteria, while pointing out possible improvements.

There are three types of comments. Inline Comments are used to comment on a particular word, sentence, or figure: they will be mainly used to suggest revisions. Review Comments are attached to a rating to explain and justify it: they correspond to the textual component in current peer reviews. Global comments are displayed in a contribution's discussion page to foster exchange of opinions. A 4th element in Mimosa is that Review Comments themselves can be scored by the community, to identify good reviewers and bad reviewers: this aspect is absent from the current paper system, causing issues of fairness and quality of reviews.

All the ratings of a contribution are weighted before aggregation: a rating has a higher weight if the rater has a high reviewer score, and ratings of newer versions of the contribution are weighted higher than older ratings.

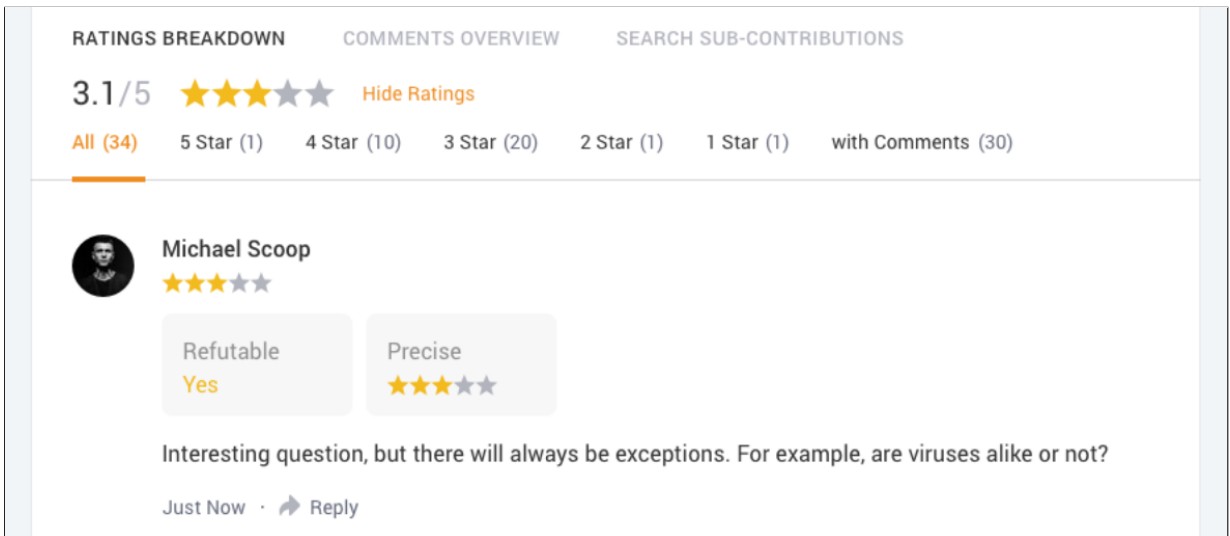

The main protection of Mimosa against abuses of the rating function will be its Reviewer Score reputation system, so that ratings from a newer account cannot influence the global ratings of the contribution, until that account gains a positive Reviewer Score. Accounts with no reputation can still have their comments voted up, so that a fair criticism by a new account can still influence the community.

A user can choose anonymity when posting a rating and Review Comment. The username will not be displayed, but the score of the Review Comment will still be aggregated in the user's Reviewer Score. These measures protect vulnerable reviewers against retaliation, while keeping consequences for bad reviewing behavior, two major issues of the current paper review process, whether single blind, double blind, or open.

# Bibliometrics

## Impact metrics

The (numerical) impact of a contribution in Mimosa is proportional to how well it is rated by the community, and to the number of sub-contributions that are appended to it. For example, an hypothesis that generates interest might cause many users to append experiments to that hypothesis, and the experimental data might be analysed by several users independently. If highly rated analyses have the same conclusion (support or refute the hypothesis), this hypothesis will weigh heavily in the final answer to the question. This system has many advantages, including the fact that null results matter and can be highly rated by the community, and that contributions that are cited as bad examples or because the author is famous do not have higher bibliometrics measures since there is no citation count.

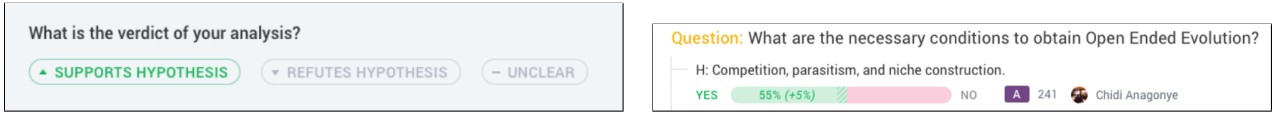

## Author metrics

Citation count and h-index are not integrated into mimosa. Contributions (and by extension their authors) are rated by the community on their fine grained characteristics. Authors with high rated contributions receive badges, for example badges for high scores in hypothesis clarity, experimental protocol replicability, or analysis thoroughness.

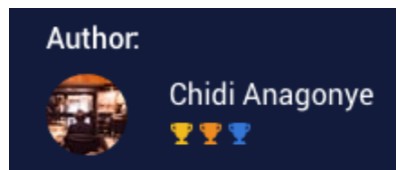

## Deprecation

When Mimosa contributions are cited in old school papers, the score of the cited contribution can be integrated in the PDF bibliography. Contributions can also be embedded in dynamic content (blogs, news, etc), in which case the current status and score of the contribution will be kept up to date. It is possible to input from other formats to mimosa by hand, but hard to import from mimosa into classical formats, as a contribution's author list might be extremely long.

## Durability

Mimosa will be open code and open data, except for private pages. This should hopefully inspire people to create their own Mimosa-like platforms, cross-operable because sharing the same ORCID-based login, which would lessen the impact of a platform dying. Unfortunately this means that the

survival of the platform depends on people's goodwill. Mimosa will eventually be funded through taking a percentage of monetary patronage of projects and individual scientists, in a format similar to Patreon/Kickstarter, except that Mimosa should remain nonprofit.

# Similar projects

Information about similar platforms can be difficult to gather, especially for projects that have been abandoned. Here are some initiatives that are currently "alive", and a their differences with Mimosa:

## Hypergraph

Hypergraph (Chris Hartgerink 2021) is a project allowing researchers to keep track of their research step by step, and share it with their followers using different formats. Hypergraph increases transparency by encouraging researchers to publish their research path, decreasing the risk of post-hoc explanations. From the Hypergraph website:

*In Hypergraph, you only have to indicate what step you're taking, link it to the step(s) it follows from, add relevant files, and indicate what file you want people to see first. This means that you can share all kinds of steps in your research—the theory, predictions, transcripts, materials, code, data, results (and more). Plus, you can share files such as Jupyter notebooks, scripts, data files, videos, audio files, text, and any other open file format.*

Hypergraph and Mimosa share their dedication to the publish-as-you-go model of publication, and to the goal of facilitating building upon others' research. The main conceptual difference between Hypergraph and Mimosa is their focus: Hypergraph focuses on highlighting the temporal development of research projects, while Mimosa only integrates basic versioning. Mimosa puts more weight on reviewing and commenting on others' research outputs, functions that do not seem to be in Hypergraph.

Hypergraph and Mimosa also have major implementation differences. Mimosa is web-based, public and centralised, while Hypergraph is decentralised, runs on the user's computer, and outputs are not public but shared within the users' followers.

## Octopus

Octopus (Alexandra Freeman 2021) is the project most similar to Mimosa. It shares the micro-publication and publish-as-you-go model, and also has reviewing functions. There are conceptual differences: Octopus publications cannot be standalone and must all be linked to another publication, and most importantly, Octopus does not aim to be a crowdsourced science platform ("I definitely wouldn't describe Octopus as a crowdsourced science platform", Alexandra Freedman, 2020, private communication), which is the main aim of Mimosa.

## Micro-publications

Some platforms are exclusively dedicated to collecting micro-publications and seem to be doing well, without the other functions (linked publications, chronological steps...) of the platforms described above. microPublication (*microPublication* 2021) is peer reviewed and focuses on biology; The Journal of Brief Ideas (David Harris, Arfon Smith, Stuart Lynn, Lars Holm Nielsen 2021) is more general and not peer reviewed. Although even traditional publishers often do not explicitly forbid short-format submissions, these two examples show that there is a demand and a public for explicitly different publication formats.

# Accessibility statement

Mimosa lowers the barrier for non-scientists to consume and contribute to science. The platform will conform to the WCAG2.1 guidelines at level A, as required for websites published by Sony employees. It will also aim for 0 errors with the Website Accessibility Evaluation Tool (*Website Accessibility Evaluation Tool* 2021) for the static content of the pages. For user-generated content, users will be requested to check that the content they share (such as images and videos) has basic accessibility: closed captions for videos, alt text and color guidelines for images.

As a commitment to accessibility from the start rather than an afterthought, this pdf document contains alt text for images, and a design description of the website mock-up for the visually impaired is available at https://bit.ly/3sZDHME.

# Limitations

## Mimosa's role in the Science ecosystem

Mimosa's enforced structure of question-hypothesis-experiment is not well suited to science that is not hypothesis driven, opinion papers, and literature reviews.

## Open problems

### Ownership-based incentives

Mimosa encourages massive collaboration, which complicates questions of research ownership. Being the first, last, or only author of a paper is currently highly socially valued, especially for job hunting in a saturated market. Mimosa is not well suited to this method of evaluating a job candidate; instead, the platform assumes that community-valued skills are important. In the current research job market, a candidate who exclusively uses Mimosa will be at a strong disadvantage. This lowers the incentive to use the platform as long as it is not recognized by major players.

### Moderation

The platform will use preemptive moderation: before publication, contributions and comments must be vetted by random members of the community, to check for respect of community standards regarding hate speech and verbal abuse. Issues that go beyond that, such as unethical research or fabricated data, will only be flagged after publication. This is a problem that would hopefully be better handled on Mimosa than traditional papers, since Mimosa's contents are not static. But it will still be a recurring issue.

### Influence networks

Mimosa will also have to face the risk of sock puppet accounts, bot networks, and "rich get richer" social circles of influential scientists, who could all contribute to bias the ratings and perception of user contributions. Some of these issues exist in journals as well, and some will be particular to this type of platform.

# References and Citations

Alexandra Freeman (2021). *Octopus*. https://science-octopus.org/. Online; accessed 12 April 2021.

Carter, K Codell (1985). "Ignaz Semmelweis, Carl Mayrhofer, and the rise of germ theory". In: *Medical History* 29.1, pp. 33–53.

Chris Hartgerink (2021). *Hypergraph*. https://www.libscie.org/hypergraph. Online; accessed 12 April 2021.

David Harris, Arfon Smith, Stuart Lynn, Lars Holm Nielsen (2021). *The Journal of Brief Ideas*. https://beta.briefideas.org/. Online; accessed 12 April 2021.

Kidwell, Mallory C et al. (2016). "Badges to acknowledge open practices: A simple, low-cost, effective method for increasing transparency". In: *PLoS biology* 14.5, e1002456.

*microPublication* (2021). https://www.micropublication.org. Online; accessed 12 April 2021.

*Website Accessibility Evaluation Tool* (2021). https://wave.webaim.org/. Online; accessed 15 March 2021.

