# OpenReview forum: "The Mimosa Manifesto: a Web Platform for Open Collaboration in Science"
_ICLR.cc/2021/Workshop/Rethinking_ML_Papers/Exhibit_and_Workflow — Rethinking ML Papers - ICLR 2021 workshop Poster_

### Official Review · Reviewer_N5CK · 2021-03-25
**Well described, but more information could be provided**

**Accessibility:**

Score of 4 (Strong): Submission states accessibility concerns and provides solutions within the proposed framework. However, it does not declare the limitations and exceptions.

**Groundsforrejection:**

This work does break double blind, which may not be a concern to the organizers.

Additionally, this work does not address the specific research needs of machine learning researchers, which suggests that this is a more general science solution. This limitation could be considered not to be an issue however by the organizers.

**Litreview:**

Score of 3 (Neutral): The submission acknowledges previous work, but does not necessarily explain how the submission differentiates itself (i.e we want to avoid the “deluge of citation” strategy, leaving the reviewer to click through references and figure this part out for themselves).

**Problemstatement:**

Score of 4 (Strong): The submission sets a very strong example of how to address the problem, which should be relevant to the workshop themes.

**Relevance:**

Score of 3 (Neutral): Attempt was clearly made to address a theme of the workshop, but it seems that the work was ‘retrofitted’ to match the theme of the workshop.

**Results:**

Score of 3 (Neutral): Submission is well designed and provides a good level of coherency/novelty/interactivity.

**Reviewerconfidence:**

Based on my professional experience, research, and technical contributions in open science, I am highly confident.

**Reviewtext:**

The author describes a website for collaborative open science, Mimosa.  On Mimosa, open discussions of research hypotheses, and questions are shared through micro-publications.  The author notes how different audiences such as students and journalists can benefit from using Mimosa.  They also describe how quality communication and moderation are maintained on the site.

Some limitations:

- Few citations or comparisons to other methods of sharing micro publications eg:

https://osf.io/
Academic Twitter
https://www.micropublication.org/
arxiv

- The website presented is only a mockup, so actual interaction is limited

- While some accessibility is described for various audiences, often accessibility is framed also in terms of providing better design for users with disabilities, which I don't see addressed

- Content moderation seems to be an important aspect of getting this website to function well, which is discussed, but does not address concerns regarding online harassment and bad actors. Given the fact that social media platforms have had limited responses to harassment of academics for expressing polarizing views, I believe that this area is in need of additional consideration to ensure quality discussions.

- More broadly, as this workshop is geared towards machine learning,  I am concerned by the fact that the authors don't address how machine learning researchers would specifically benefit from this website.  To me, this does suggest that this work is meant to be a website for all sciences and that machine learning researchers needs are not particularly considered.  This choice in presentation and description however limits its applicability to the workshop.

I think this would be an interesting work to be presented and would like to see a full prototype at the conference. I believe that an actual working prototype would best engage the workshop community.


**Score:**

Accept: The reviewer believes the submission provides a novel and reliable scheme to improve science communication but needs improvement.

---

> ### Author Response · Authors · 2021-05-16
> **Revised literature review and future work**
>
> I would like to thank the reviewer for their feedback, and answer to some of the comments.
>
> > Few citations or comparisons to other methods of sharing micro publications
> I have added comparisons to similar initiatives: Octopus and Hypergraph.
>
> > The website presented is only a mockup
> This is a fair point: at the time of submission the actual website was still in construction. That being said, the mockup was as complete as possible and included all possible interactions types.
>
> > often accessibility is framed also in terms of providing better design for users with disabilities, which I don't see addressed
> The submission included an accessibility statement pledging to conform to the WCAG2.1 guidelines at level A (which is geared towards users with disabilities), a link to a full textual description of the prototype created by an accessibility expert for Blind users, and the submission itself included screen-reader friendly alt-text for every image in the PDF.
>
> > does not address concerns regarding online harassment and bad actors.
> Although not discussed in the submission, this is definitely a main concern. For now the plan is to review every submission for violations of the conduct policy before publication, but I must find a scalable way to address the issue...

---

### Official Review · Reviewer_bqvT · 2021-03-31
**Interesting Alternative to Traditional Publications**

**Accessibility:**

Score of 5 (Exceptional): Submission identifies and articulates accessibility matters, provides justifications for the proposed paradigm, and declares the limitations.

**Litreview:**

Score of 2 (Needs Improvement): The submission leaves out prominent examples of previous work in the area.

**Problemstatement:**

Score of 4 (Strong): The submission sets a very strong example of how to address the problem, which should be relevant to the workshop themes.

**Relevance:**

Score of 4 (Strong): The submission directly addresses a theme of the workshop, and does so in a very professional manner.

**Results:**

Score of 3 (Neutral): Submission is well designed and provides a good level of coherency/novelty/interactivity.

**Reviewerconfidence:**

3. I believe the proposed Mimosa platform could be a valuable tool to supplement traditional conference/journal and gain early feedback from active researchers.

**Reviewtext:**

The paper offers an alternative take on traditional publishing through their proposed Mimosa platform. The idea solves some of the issues faced by conferences and journals (e.g., long wait times, non-collaborative environment) by allowing users to create research posts and updates that other researchers can comment on and add results to. In principle, this is a nice way to get early feedback and collaborate with active researchers in your field.

As the author noted, there are some downsides to this form of open-research, including the lack ownership-based incentives which complicates the question of "research ownership", an important metric for jobs, promotion etc. However, Mimosa could be used as a supplementary platform to collaborate and get early feedback, not necessary as a sole replacement for conferences/journals.

While I'm not an expert in open-research platforms, I feel it is likely there is more related research than the 2 citations provided. Even if not a directly related to the proposed Mimosa platform, there are works that discuss new forms of scientific communication, and a comparison of proposed alternatives would be nice.  A quick search revealed:

(Micro)Blogging Science? Notes on Potentials and Constraints of New Forms of Scholarly Communication by Cornelius Puschmann

**Score:**

Accept: The reviewer believes the submission provides a novel and reliable scheme to improve science communication but needs improvement.

---

> ### Author Response · Authors · 2021-05-05
> **Related research**
>
> I would like to thank the reviewer for their feedback.
>
> Indeed there is more related research and even related platforms, and I added a section to the manuscript to reflect that.
> In reality the manuscript was initially planned to be a short manifesto rather than a paper, but it ended up being an awkward length that satisfy neither formats and lacks some of the rigor of a paper with proper literature review... I will continue to include additional references in the coming days.

---

### Official Review · Reviewer_vaNN · 2021-03-31
**Maybe over too broad, but well-written and worth discussing**

**Accessibility:**

Score of 4 (Strong): Submission states accessibility concerns and provides solutions within the proposed framework. However, it does not declare the limitations and exceptions.

**Groundsforrejection:**

Cannot think of any

**Litreview:**

Score of 4 (Strong): The submission directly differentiates itself from previous works and formats.

**Problemstatement:**

Score of 5 (Exceptional): The submission states a well-known problem relevant to the workshop, and sets what could be a new standard in the field when it comes to addressing it.

**Relevance:**

Score of 4 (Strong): The submission directly addresses a theme of the workshop, and does so in a very professional manner.

**Results:**

Score of 5 (Exceptional): Submission has an excellent design and all criteria are addressed. Conclusions, practical/theoretical implications are well articulated.

**Reviewerconfidence:**

4

**Reviewtext:**

The paper proposes a radically new crowdsource and micro-contribution-based way of doing research and disseminating results, called Mimosa. The paper is of that rare kind, where reader (me)  has some questions and objections after Section N, and Section (N + 1) itself addresses them by answering them or directly listing as a limitations. I had a great pleasure reading the manifest.

That being said, I am quite skeptical about Mimosa as a project, mostly because lots of ML & CV papers, including all my own papers are not of the type "question-hypothesis-experiment". I also do not believe that the statement "scientific papers are more interesting and more rigorous when they are written by two people who start out genuinely disagreeing" is generally true.
Something somewhat similar to Mimosa for ML -- open collaboration, micro-contributions,  was tried at AI-ON, an organization, founded by Bengio, some researchers from Google, Facebook, etc, (see some info here --https://github.com/AI-ON/ai-on-project-example , the main website ai-on.org   is dead now), unsuccessfully. I recommend author to learn from that failure.

Nevertheless, I believe that the presenting the Workshop as a discussion spot for Mimosa would greatly benefit the ML community, specifically one could start to rething the way of ML research is done & distributed.  The manifest is clearly in the scope of the workshop, or, to be more precise - the workshop is partly in the scope of the manifest. The rest of points (Accessibility, Problemstatement, etc) are also good.

**Score:**

Accept: The reviewer believes the submission provides a novel and reliable scheme to improve science communication but needs improvement.

---

> ### Author Response · Authors · 2021-04-30
> **The fate of AI-ON**
>
> I would like to thank the reviewer for their comments and for pointing out AI-ON, which I was able to access through the Internet Archive. Unfortunately with this project as with much of science, the reasons why it failed do not seem to have been explored publicly, so I am currently looking for people who used the platform to get their feedback.
>
> > lots of ML & CV papers, including all my own papers are not of the type "question-hypothesis-experiment".
>
> That is one of the main limitations of the platform (in my opinion); on the other hand, limiting the platform to "question-hypothesis-experiment" based research helps to reduce the aims to a manageable scope, at least during the first phases of its existence.

---

### Meta-Review · Program_Chairs · 2021-04-01

**Recommendation:** Accept
**Confidence:** 4

**Metareview:**

This paper introduces a free and open-collaborative online platform to discuss research ideas and get early feedback. Although I found this platform promising for collaborations across the field, there needs to be well-defined rules and guidelines on how to use this intra-collaborative channel in order to promote high-profile researchers' involvement and avoid any disrespectful wordings and misbehaviors that often happen in online and indirect conversations.

I second the reviewers' perspectives on providing a more suitable design for users with disabilities. As an example, blind or color-blind users need to be considered. As mentioned, the concerns of accessibility and inclusivity in Machine Learning (ML) should be addressed in Mimosa, as this has not been discussed in the current submission.

Overall, I vote for accepting this submission. Designing a generally acceptable open science platform would be a great movement in the field and worth discussing in the workshop. However, I strongly recommend the authors incorporate reviewers' feedback in their final version, such as elaborating on the previous and relevant works, presenting the ML researchers' limitations, and how Mimosa can improve upon them.

Moreover, the current submission breaks the double-blind review which had to be accommodated.

---

> ### Author Response · Authors · 2021-04-30
> **Single-blind submission as per submission guidelines**
>
> I would like to thank the meta-reviewer for their comments, and will update the submission according to reviewers comments.
>
> I would also like to point out that this submission broke the double-blind review as per the submission guidelines (https://rethinkingmlpapers.github.io/submit/), which state that  "exhibit and workflow submissions will be reviewed in a single-blind process".

---

### Decision · Program_Chairs · 2021-04-01

Accept (Poster)